# Immune Status of Cervical Lymph Nodes in Head and Neck Cancer—A Surgical Oncology Perspective

**DOI:** 10.3390/jpm13071174

**Published:** 2023-07-22

**Authors:** Hiromu Nakamura, Tetsuya Ogawa, Shunpei Yamanaka, Daisuke Inukai, Takashi Maruo, Taishi Takahara, Akira Satou, Toyonori Tsuzuki, Susumu Suzuki, Ryuzo Ueda, Yasushi Fujimoto

**Affiliations:** 1Department of Otorhinolaryngology-Head and Neck Surgery, Aichi Medical University School of Medicine, 1-1 Yazakokarimata, Nagakute 480-1195, Aichi, Japan; nakamura.hiromu.291@mail.aichi-med-u.ac.jp (H.N.); yamanaka.shunpei.036@mail.aichi-med-u.ac.jp (S.Y.); inukai.daisuke.006@mail.aichi-med-u.ac.jp (D.I.); maruo.takashi.712@mail.aichi-med-u.ac.jp (T.M.); fujimoto.yasushi.839@mail.aichi-med-u.ac.jp (Y.F.); 2Department of Surgical Pathology, Aichi Medical University School of Medicine, 1-1 Yazakokarimata, Nagakute 480-1195, Aichi, Japan; takahara.taishi.456@mail.aichi-med-u.ac.j (T.T.); satou.akira.442@mail.aichi-med-u.ac.jp (A.S.); tsuzuki@aichi-med-u.ac.jp (T.T.); 3Research Creation Support Center, Aichi Medical University School of Medicine, 1-1 Yazakokarimata, Nagakute 480-1195, Aichi, Japan; suzukis@aichi-med-u.ac.jp; 4Division of Medical Sciences, Department of Physiological Science, Nagoya University Graduate School of Medicine, Furo-cho, Chikusa-ku, Nagoya-shi 464-8601, Aichi, Japan; ueda.ryuuzou.962@mail.aichi-med-u.ac.jp

**Keywords:** head and neck cancer, neck dissection, human papillomavirus, immunotherapy, immune response, regional lymph nodes, T cells

## Abstract

Neck dissection for cervical lymph node metastasis is an established procedure for head and neck cancer (HNC). However, with the advent of immunotherapy, head and neck surgical oncologists need to rethink removing all lymph nodes, including those with immune function. We investigated the anti-cancer immune response of the cervical lymph nodes in four patients with human papillomavirus type 16 (HPV16)-positive head and neck squamous cell carcinoma. Using lymphocytes extracted from local, metastatic, and non-metastatic lymph nodes and peripheral blood from these patients, we performed an intracellular flow cytometric cytokine assay using anti-IFNγ and anti-TNF-α monoclonal antibodies to detect HPV16 E6- and E7-specific T cells. HPV status and p16 immunostaining were determined by in situ detection using the HPV RNAscope method and immunohistochemistry. In one case, E6-specific and E7-specific CD8+ T cells were detected in proximal metastatic nodes and distal non-metastatic nodes. This finding suggests that non-metastatic nodes should be preserved for their immune function during neck dissection and that the immune function of non-metastatic lymph nodes is important when administering immunotherapy. In this context, head and neck surgical oncologists treating HNC should consider the place of immunotherapy and neck dissection in the treatment of HNC.

## 1. Introduction

Head and neck cancer (HNC) is the seventh most common type of cancer, with oral cancer being the most prevalent, followed by pharyngeal cancer. Histologically, squamous cell carcinoma accounts for about 90% of HNCs, and lymph node metastasis at the time of initial diagnosis can occur in as many as 40% of cases [1]. Neck dissection, the definitive treatment for lymph node metastasis, was first proposed by Crile in 1906 [2], and radical neck dissection was subsequently established by Martin et al. [3]. Neck dissection is based on Halstead’s theory that en bloc resection is important in the surgical treatment of cancer, with radical resection of the primary site, thorough regional dissection, and removal of local and surrounding cervical lymph node tissue being key to improving the cure rate [4]. More function-preserving techniques were subsequently proposed. The neck was divided into 5 anatomical regions, and selective neck dissection to reduce the dissected area and modified neck dissection to reduce functional disability are the neck dissection techniques currently used [5]. However, Halstead’s theory of en bloc resection is still accepted.

Meanwhile, immunotherapeutic approaches for head and neck squamous cell carcinoma (HNSCC) have been developing rapidly following the advent of immune checkpoint inhibitor (ICI) therapy using the anti-programmed death (PD)-1 therapeutic antibodies nivolumab [6] and pembrolizumab [7].

Anti-cancer T cell responses are amplified by stepwise events in the 7-step cancer immunity cycle [8]. In the third step of this cycle, T cells are primed with dendritic cells presenting cancer antigens in regional lymph nodes, and it was recently reported that ablation of these lymph nodes eliminated the effect of ICI therapy in an experimental murine model of HNSCC [9]. Therefore, regional lymph nodes are crucial for activating immunity against cancer in HNSCC. At present, surgery is generally the treatment of choice for resectable HNSCC, and ICIs are often used after neck dissection with en bloc resection of the cervical lymph nodes. However, neck dissection involves the removal of both metastatic lymph nodes (MLN) and non-MLN, and if non-MLN have anti-cancer immune activity, the concept of neck dissection is incompatible with that of recent immunotherapy. For these reasons, surgical oncologists must determine the impact of neck dissection on the efficacy of immunotherapy in HNSCC by examining the anti-tumor immune response in the MLN and regional lymph nodes.

Human papillomavirus (HPV) has been identified as a major cause of oropharyngeal carcinoma (OPC), and approximately half of all patients with OPC in Western countries are HPV-positive [10]. Most cases of HPV-positive HNSCC are reported to result from high-risk HPV type 16 (HPV16). High-risk HPV DNA is integrated into the genome and overexpresses the viral oncogenes E6 and E7. E6 binds to p53, a regulatory protein in the cell cycle, leading to its degradation, while E7 binds to the tumor suppressor Rb and causes its degradation, resulting in overexpression of p16. A combination of these molecular abnormalities leads to the uncontrolled growth and proliferation of tumor cells [10,11]. E6 and E7 HPV proteins not only have oncogenic activity but also display high antigenicity. High titers of serum antibodies against E6 and E7 have been detected in HPV-positive HNSCC, and HPV-specific T cells are associated with the prognosis [12]. Vaccines that target E6 and E7 have now been developed to prevent primary HPV infection [12,13,14], and the detection of the immune response to HPV E6 and E7 can be used to evaluate anti-cancer activity in patients with OPC. Therefore, in this study, we used overlapping peptide pools (OPP) derived from the HPV16 proteins E6 and E7 obtained from surgical specimens in 4 patients with p16-positive oropharyngeal carcinoma who underwent local resection and neck dissection to examine the local, metastatic, proximal, and distal lymph nodes and peripheral blood for CD8+ T cell and CD4+ T cell immune responses measured in vitro.

## 2. Materials and Methods

### 2.1. Patient Selection

For this study, we selected a total of four patients who had undergone neck dissection and primary site removal at the Department of Otorhinolaryngology-Head and Neck Surgery, Aichi Medical University Hospital, within the period of 2020 to 2022. Out of the four patients, three were diagnosed with oropharyngeal carcinoma (OPC), and one had squamous cell carcinoma with an unknown primary site and metastasis to the lymph nodes. To ensure ethical standards, all patients included in the study were enrolled with the approval of the Ethics Committee of Aichi Medical University (approval number 2020-H033). The study adhered to the guidelines and regulations set by the committee to safeguard patient rights and welfare. The distribution of lymph nodes was determined based on the standardized classification system recommended by the American Academy of Otolaryngology—Head and Neck Surgery. This classification system provides a consistent and widely accepted framework for identifying and categorizing lymph nodes in the head and neck region. By following this standardized system, we aimed to ensure consistency and comparability in our study’s findings. The selection of these patients and adherence to ethical guidelines, along with the utilization of a standardized classification system for lymph node distribution, contribute to the reliability and validity of our study. These measures enable us to analyze and interpret the results accurately, providing valuable insights into the immune response to HPV-specific antigens in HNC patients.

### 2.2. Immunohistochemistry

Immunohistochemical staining was employed to detect the presence of p16. To begin, tissue sections measuring 3 µm in thickness were prepared from HNSCC samples. The slides underwent deparaffinization and rehydration processes before being immersed in Tris-EDTA buffer at a pH of 9.0 to facilitate antigen retrieval. Subsequently, a 3% H_2_O_2_ solution was applied to the slides for a duration of 10 min to block endogenous peroxidase activity. In order to minimize non-specific binding, the sections were then blocked for 1 h with 5% goat serum. Following this, the primary antibody specific to p16 was applied to the sections and allowed to incubate overnight at a temperature of 4 °C, ensuring specific binding of the antibody to the target antigen. To visualize the presence of p16, the slides underwent a two-step method using immunohistochemical reagents, which involved the application of secondary antibodies that recognize the primary antibody-antigen complex. This was followed by treatment with a 3,3-diaminobenzidine solution, resulting in the formation of a brown-colored precipitate at the site of antigen-antibody binding. To assess the staining pattern and distribution of p16, ten randomly selected high-power fields were observed under a BX51 microscope (Olympus, Tokyo, Japan) equipped with a 20× objective. This microscopic examination allowed for the evaluation of p16 expression and localization within the HNSCC tissue samples. By performing immunohistochemical staining for p16, it becomes possible to analyze its presence and distribution, providing valuable insights into its potential role as a diagnostic or prognostic marker in HNSCC. The examination of multiple high-power fields ensures representative sampling and a comprehensive assessment of p16 expression patterns in the tissue sections.

### 2.3. HPV16 mRNA In Situ Hybridization

To investigate the presence of HPV, in situ hybridization was performed using the RNAscope method. This technique utilizes specific probes that target and bind to the mRNA of interest. The scoring system employed for the interpretation of the results is as follows: Score 0: No staining or fewer than one dot observed in every 10 cells, visible at a magnification of 40×.Score 1: 1–3 dots per cell, visible at a magnification of 20–40×.Score 2: 4–10 dots per cell, with minimal dot clustering, visible at a magnification of 20–40×.Score 3: More than 10 dots per cell, with less than 10% of positive cells showing dot clusters, visible at a magnification of 20×.Score 4: More than 10 dots per cell, with more than 10% of positive cells displaying dot clusters, visible at a magnification of 20×.Cases with an RNAscope score equal to or greater than 1 were considered positive for HPV. This scoring system allows for the quantification and characterization of HPV-specific mRNA signals within the tissue samples. By evaluating the number of dots per cell and the presence of dot clusters, the scoring system provides a standardized approach to determining the level of HPV mRNA expression in the examined cells. Positive cases indicate the presence of HPV mRNA, suggesting an active HPV infection or viral gene expression within the tissue samples.

### 2.4. Lymphocyte Preparation and Detection of HPV16 E6- and E7-Specific T Cells

To increase the number of HPV-specific T cells to a detectable level, the lymphocytes were re-suspended in Roswell Park Memorial Institute 1640 medium supplemented with 10% fetal bovine serum. They were then stimulated with oligopeptide pools (OPP) derived from HPV-encoded antigens E6 and E7. (E6 peptide; PepTivator^®^HPV16E6-premium grade, HPV16 E6 peptide 15-mer sequences with 11 amino acids overlap (50 peptides); E7 peptide; PepTivator^®^HPV16E7-premium grade, HPV16-E7 peptide 15-mer sequences with 11 amino acids overlap (21 peptides)).

The purpose of this stimulation was to activate the T cells specific to these antigens. The stimulated lymphocytes were cultured for a duration of 14 days in the presence of interleukin-2, a cytokine known to support T cell growth and proliferation. Following the culture period, both the in vitro-expanded T cells and non-cultured lymphocytes were subjected to staining procedures for further analysis. The staining process involved the use of specific monoclonal antibodies, including BUV737-CD3, BV605-CD4, and BUV395-CD8. These antibodies were utilized to identify and distinguish different subsets of T cells based on their surface markers. The staining procedure was carried out for a duration of 20 min at a temperature of 4 °C. After the staining process, the cells were fixed with 20% formalin for an additional 20 min at 4 °C. To ensure optimal staining and minimize non-specific binding, the cells underwent two washes in phosphate-buffered saline containing 0.2% human serum albumin and 2-mM EDTA. The next step involved staining the cells with fluorescein isothiocyanate-interferon gamma (IFN-γ) and activated protein C-tumor necrosis factor (TNF)-α monoclonal antibodies. These antibodies were diluted 20-fold in phosphate-buffered saline supplemented with saponin and bovine serum albumin. The staining process was carried out for 30 min at 4 °C. To detect specific T cell responses to HPV antigen, IFNγ and TNFα produced in T cells by OPP stimulation were detected using a Fortessa X20 flow cytometer (BD Biosciences, Franklin Lakes, NJ, USA). This instrument allows for the precise measurement and analysis of fluorescence signals emitted by the stained cells. The acquired data were further analyzed using FlowJo software (Tree Star, Ashland, OR, USA), which enables the characterization and quantification of T cell subsets based on their antigen-specific reactivity. By employing these laboratory techniques, the researchers were able to evaluate and quantify the immune response of HPV-specific T cells in lymphocyte samples obtained from different anatomical sites. This comprehensive analysis provided valuable insights into the presence and distribution of HPV-specific T cells in patients with head and neck cancer.

## 3. Results

### 3.1. Patient Characteristics and Pathology

The patient with the unknown primary (case 1) presented with T0N1M0 disease, indicating the absence of a detectable primary tumor but the presence of metastasis in one lymph node. In this case, the lymph node containing metastatic squamous cell carcinoma was specifically identified at level I (pT0N1M0). Among the three patients diagnosed with oropharyngeal carcinoma (OPC), case 2 was classified as having T1N1M0 disease. This signifies the presence of a primary lesion on the anterior wall of the oropharynx with lymph node metastasis at level II (pT1N1M0). The primary tumor was limited in size but had spread to a nearby lymph node. Cases 3 and 4, both categorized as pT2N1M0, exhibited primary lesions in the lateral wall of the oropharynx. Lymph node metastasis was observed at level II, indicating the spread of cancer cells to regional lymph nodes. In both cases, the primary tumors were larger in size compared to case 2. During the surgical interventions, non-metastatic lymph nodes located at both proximal and distal sites were carefully dissected and removed in cases 3 and 4. These non-metastatic lymph nodes, referred to as non-MLN (non-metastatic lymph nodes), were included in the analysis. The removal of these non-MLNs allows for a comprehensive evaluation of the lymphatic system and aids in understanding the immune response and potential metastatic patterns associated with oropharyngeal cancer (Table 1).

### 3.2. Immunohistochemistry of p16

Immunostaining results revealed positive staining in all cases, as shown in Figure 1.

### 3.3. HPV Status Determined by RNAscope Targeting HPV E6 and E7 mRNA

The application of HPV in situ hybridization using the RNAscope system resulted in positive outcomes across all four cases, as illustrated in Figure 2.

### 3.4. HPV16 E6- and E7-Specific T Cells

In the three patients with oropharyngeal carcinoma (cases 2–4), the immune response from HPV16 E6-specific or E7-specific T cells was not observed at any of the sampled sites. However, in case 1, a robust immune response was demonstrated by the presence of HPV16 E6- and E7-specific T cells in both the lymphocytes from peripheral blood and the lymph nodes. Figure 3A illustrates the positivity rates of different T cell subsets based on their production of interferon-gamma (IFN-γ) and tumor necrosis factor-alpha (TNF-α). In the MLN, the rates of IFN-γ+/TNF-α+, IFN-γ+/TNF-α-, and IFN-γ-/TNF-α+ subsets were 0.51%, 1.31%, and 0.73%, respectively. Remarkably, these rates were approximately doubled when the MLN were restimulated with E6 peptides compared to when they were not restimulated. In the DLN, only the IFN-γ-/TNF-α+ subset exhibited a positive immune response. When restimulated with E6 oligopeptide pools (OPP), the positivity rate of this subset was 1.03%, approximately three times higher than that observed without restimulation. However, no positive reaction was detected in any subset of peripheral blood lymphocytes (PBL) or proximal lymph nodes (PLN). Notably, HPV16 E6-specific CD8+ T cells were specifically identified in both the MLN and DLN. These findings provide important insights into the immune response against HPV16 E6 and E7 antigens in head and neck cancer patients. The data suggest that there is a site-specific immune response in lymph nodes, particularly in MLN and DLN, with a higher prevalence of HPV-specific T cells. The presence of HPV-specific T cells in these lymph nodes indicates their potential role in the immune defense against HPV-associated malignancies. However, it is worth noting that the immune response in this study was limited to a single case with a positive response, highlighting the need for further investigations with larger patient cohorts to validate and generalize these findings.

Furthermore, upon restimulation of the cells with E7 peptides, the positive rates in each fraction demonstrated notable increases. In the MLN, the positive rates for the IFN-γ+/TNF-α+, IFN-γ+/TNF-α-, and IFN-γ-/TNF-α+ subsets were 4.38%, 4.89%, and 1.18%, respectively, representing a significant enhancement compared to the non-restimulated conditions (Figure 3B). Similarly, in the PLN, the positive rates for these subsets were 2.92%, 5.44%, and 2.78%, respectively, showing a remarkable 2- to 60-fold increase compared to the non-restimulated state. Notably, no positive reaction was observed in any subset of PBL or DLN. HPV16 E7-specific CD8+ T cells were specifically detected in both the MLN and PLN. However, neither HPV16 E6-specific nor E7-specific CD4+ T cells were detected at any other site (Figure 3C,D). To summarize the detection levels of HPV16 antigen-specific T cells at each site in each of the four patients, Table 2 provides an overview. These findings underscore the selective activation and expansion of HPV16 E7-specific CD8+ T cells in the MLN and PLN following restimulation. The absence of detectable HPV-specific CD4+ T cell responses suggests a potential difference in the immune recognition and targeting of HPV16 antigens by CD8+ and CD4+ T cell subsets. It is important to note that these results demonstrate the individual variations in the immune response among the patients examined. Further studies with a larger patient population are warranted to validate these findings and explore the clinical implications of HPV-specific T cell responses in the context of head and neck cancer immunotherapy.

## 4. Discussion

The main finding of this study was that HPV16 E6- and E7-specific T cells showed an immune response in not only MLN but also non-MLN in one case of HNSCC. As far as we are aware, this is the first report of an immune response of non-MLN to cancer antigens in HNSCC.

Immunotherapy is rapidly becoming a standard treatment for HNC, and in view of the important role of lymph nodes in anti-cancer immunity, head and neck surgical oncologists are now required to adapt their management. Function-preserving neck dissection techniques have been proposed, but Halstead’s theory of en bloc resection still prevails. Therefore, we need to reconsider neck dissection from an immunity point of view.

At present, ICIs are rarely used first-line, and surgery is usually the treatment of choice for resectable HNSCC. However, ICIs are often used after neck dissection with en bloc resection of the cervical lymph nodes. Current neck dissection entails removing both MLN and non-MLN, and if non-MLN have anti-cancer immune activity, our current concepts of neck dissection and immunotherapy are not compatible. Therefore, head and neck surgical oncologists need to investigate the impact of neck dissection on the efficacy of immunotherapy in HNSCC by examining the anti-tumor immune response of the MLN and regional lymph nodes.

In this study, an immune response to HPV16 E6- and E7-specific T cells was found not only in MLN but also in non-MLN in one of four patients who were positive for p16 and HPV16. No immune response was seen in PBL or tumor-infiltrating lymphocytes in any of the 4 cases. There has been a report of a study in which 77% and 44% of T cells in PBL in patients with HPV16+ HNSCC were found to be HPV16 E6- and E7-specific, respectively [15]. Although detectable frequencies of HPV16 E6- and E7-specific T cells in PBL vary from report to report, possibly reflecting racial differences in HLA haplotype, the frequencies in our study were relatively low. Nevertheless, the fact that an immune response was observed in non-MLN, albeit at a low frequency, indicates the need to alter surgical treatment for HNSCC now that immunotherapy is becoming more widespread. One strategy might be to use neoadjuvant ICI chemotherapy.

Although excellent clinical effects were achieved by ICI therapy in patients with incurable HNSCC in the Checkmate-141, Keynote-040, and Keynote-048 trials, the overall survival rate at 12 months was 20–30%, which is far from satisfactory [6,7,16]. At present, ICIs are used after neck dissection, radiotherapy, or chemotherapy. We suspect that better results might be achieved if ICIs are used as neoadjuvant therapy. A recent clinical trial of neoadjuvant pembrolizumab for resectable HNSCC showed a pathological response in 44% of 36 patients without severe (grade 3–4) adverse events or unexpected surgical delays [6]. This finding indicates that neoadjuvant ICI therapy could be worthwhile.

The second option, namely, DLN preservation surgery, is attractive but very difficult to perform given the risk of residual cancer cells in the lymph nodes. Neck dissection is standard in current surgical treatment for HNSCC but is now thought to have a significant negative impact on anti-cancer immunity. Most previous studies of cancer treatment were focused on preventing the dissemination of metastases and removing non-MLN without considering the need to preserve the anti-cancer immune response. However, several experimental studies in mice have demonstrated the pivotal role of non-MLN in PD-1/PD-L1 immunotherapy [17], and clinical investigations are needed in the future. Although the significance of non-MLN preservation is widely recognized, it is impossible to determine each and every lymph node based on intraoperative findings. The topological relationship between non-MLN and the anti-cancer immune response should be investigated in frozen and fresh non-MLN samples obtained during neck resection to determine whether non-MLN should be preserved. In our study, anti-HPV16 E6 and E7 responses were higher in MLN than in non-MLN in case 1. Therefore, it seems that it is not as simple as removing only the MLN.

This study had some limitations in that it included only 4 cases and only 2 of the 7 proteins defined as HPV16 virus-derived tumor-associated antigens (E2, E4, E5, E6, E7, L1, and L2) were used to evaluate the T cell immune response. Only E6 and E7 were investigated because they are known to be oncogenic and are the best studied in terms of the HPV16 immune response. However, E2, E4, and E5 are reportedly more immunogenic than E6 or E7 [13]. Therefore, the immune response to HPV16 antigens in HPV16+ HNSCC requires further evaluation. Further studies that include cases with the other HPV16 antigens, in particular E2, E4, and E5, as well as E6 and E7, are needed for a more detailed evaluation of the immune response of DLN in patients with HPV16 + HNSCC.

## 5. Conclusions

In the present study, HPV-specific immune responses in surgical specimens collected from cervical lymph nodes were examined, and E6-specific and E7-specific CD8+ T cells were detected in proximal and distal non-MLNs as well as MLNs in some cases. This finding indicates that non-MLNs have immune function and that the immune function of non-metastatic lymph nodes is important when immunotherapy is administered. In this regard, head and neck surgical oncologists treating HNC should consider the place of immunotherapy and neck dissection in the treatment of head and neck cancer.

## Figures and Tables

**Figure 1 jpm-13-01174-f001:**
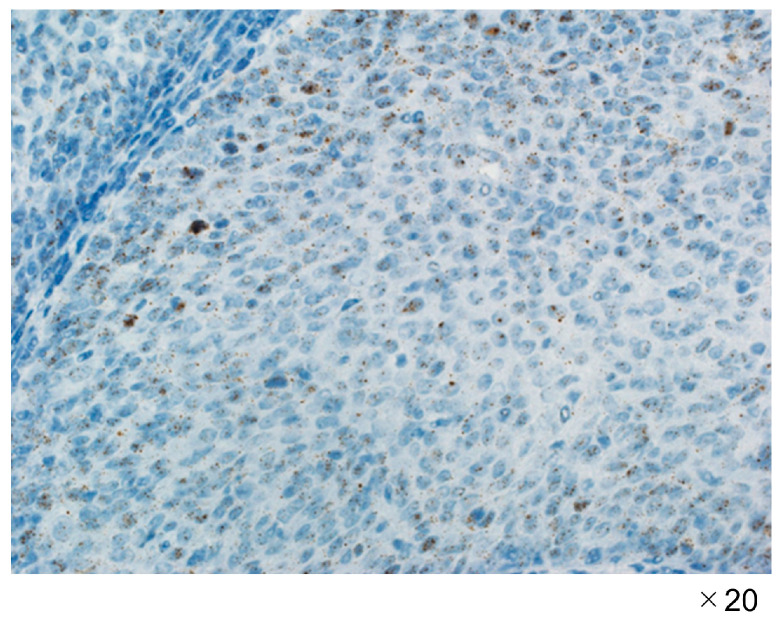
p16-Immunostained micrographs in case 1.

**Figure 2 jpm-13-01174-f002:**
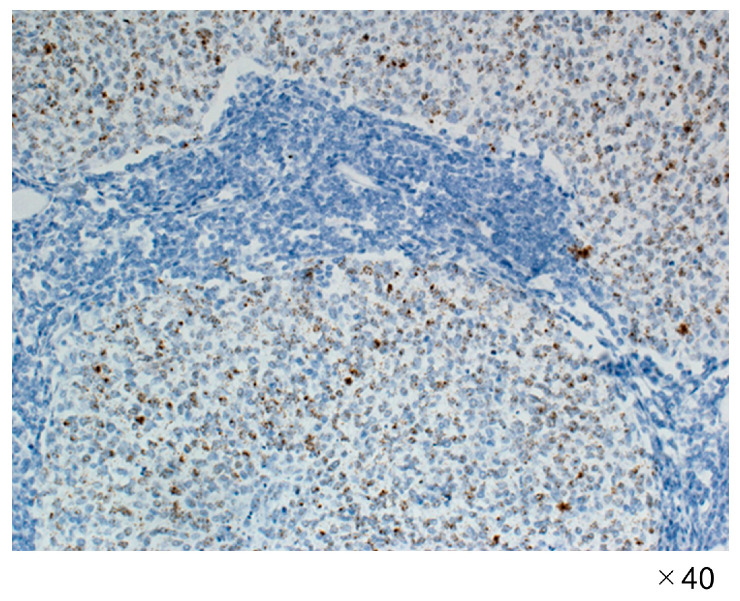
RNAscope targeting human papillomavirus type 16 E6 and E7 mRNA.

**Figure 3 jpm-13-01174-f003:**
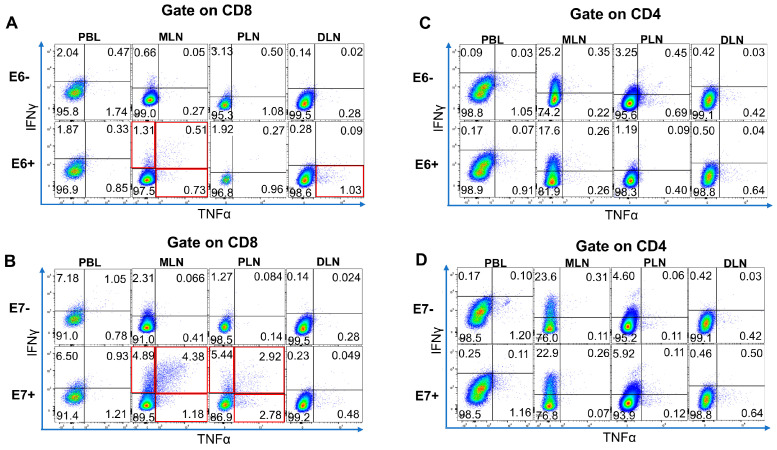
Detection of HPV16 E6-specific and E7-specific T cells in lymphocytes from peripheral blood and lymph nodes in case 1. PBL, MLN, PLN, and DLN were co-cultured with HPV16 E6 or E7 overlapping peptide pools (OPP) for 2 weeks and then restimulated with the cognate peptides. Next, intracellular IFN-γ and TNF-α were detected with flow cytometry. The cells gated on the CD8+ fraction or CD4+ fraction were separated into 2 dimensions with IFN-γ (vertical axis) and TNF-α (horizontal axis) and divided into quadrants. E6+ and E7+ indicate the cells that were restimulated with each OPP, while E6- and E7- indicate the cells that were not restimulated with any peptides. The numbers in each cytogram indicate the frequency in each quadrant as a percentage. In the event that the IFN-γ and/or TNF-α positive frequencies in restimulated cells were more than twice those in non-restimulated cells, the reaction to restimulation was judged to be specific. Red frames indicate specific reactions. (**A**) HPV16 E6-specific CD8+ T cells are detected in MLN and DLN. (**B**) HPV16 E7-specific CD8+ T cells detected in MLN and PLN. (**C**,**D**) Neither HPV16 E6-specific nor HPV16 E7-specific CD4+ T cells were detected in cells at any site. DLN, distal lymph nodes; HPV16, human papillomavirus type 16; IFN, interferon; MLN, metastatic lymph nodes; PBL, peripheral blood lymphocytes; PLN, proximal lymph nodes; TNF, tumor necrosis factor.

**Table 1 jpm-13-01174-t001:** Patient characteristics.

Patient	Primary Site	TNM Classification			MLN Level	LN		Sex	Age	HPV Status	
		T	N	M		P	D			HPV16	p16
Case 1	Unknown	0	1	0	I	II	V	M	61	+	+
Case 2	Anterior	1	1	0	II	III	V	F	43	+	+
Case 3	Lateral	2	1	0	II	III	V	F	54	+	+
Case 4	Lateral	2	1	0	II	III	V	F	60	+	+

D, distal; HPV16, human papillomavirus type 16; LN, lymph node; MLN, metastatic lymph node; P, proximal.

**Table 2 jpm-13-01174-t002:** HPV16 antigen-specific T cell detection levels at each site in the four patients.

Patient	T-Cells	Antigen	PBL	MLN	PLN	DLN	TIL
Case 1	CD4	E6	−	−	−	−	−
		E7	−	−	−	−	−
	CD8	E6	−	+	−	+w	−
		E7	−	++	++	−	−
Case 2	CD4	E6	−	−	−	−	−
		E7	−	−	−	−	−
	CD8	E6	−	−	−	−	−
		E7	−	−	−	−	−
Case 3	CD4	E6	−	−	−	−	−
		E7	−	−	−	−	−
	CD8	E6	−	−	−	−	−
		E7	−	−	−	−	−
Case 4	CD4	E6	−	−	−	−	−
		E7	−	−	−	−	−
	CD8	E6	−	−	−	−	−
		E7	−	−	−	−	−

Detection levels of intracellular cytokines (IFN-γ and/or TNF-α) are shown as follows: −, <0.5%; +w, 0.5–1%; +, 1–5%; and ++, >5%. DLN, distal lymph nodes; HPV16, human papillomavirus type 16; IFN, interferon; PBL, peripheral blood lymphocytes; MLN, metastatic lymph nodes; PLN, proximal lymph nodes; TIL, tumor-infiltrated lymphocytes; TNF, tumor necrosis factor.

## Data Availability

The data that support the findings of this study are available on request from the corresponding author. The data are not publicly available due to privacy reasons.

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
