# Peer review of "Immune Status of Cervical Lymph Nodes in Head and Neck Cancer—A Surgical Oncology Perspective"

_jpm, 2023, doi:10.3390/jpm13071174_

Round 1
Reviewer 1 Report
By analyzing in four patients with HPV16-driven head and neck cancer the activation status of T cells in peripheral blood and present in metastatic (MLN) and not disease-affected proximal (PLN) and distal lymph nodes (DLN) before and after ex-vivo stimulation utilizing a peptide pool, the authors detected in one of the four patients signs of potential response to either E6 or E7 stimulation in MLN and DLN and MLN and PLN, respectively, according to elevated numbers of CD8+ T cells staining intracellular positive for IFN-gamma and/or TNF-alpha. The good readable paper, however, has a number of limitations and a far-fetching interpretation of limited findings from intracellular cytokine staining of CD8+ T cells. Consequently, the paper requires major revisions before being suitable for publication. The main problem is that the paper based on very few data and without replication of findings in at least a second patient concludes that surgeons should abstain from resecting all lymph nodes when a neck dissection is done as there were signs for malignancy or cancer-associated inflammation. Despite that it could indeed be that keeping more tumor-antigen-specific T cells is beneficial (as suggested by some results obtained in mouse models), only partial resection of HNSCC or doing inconsequent neck surgery by removing to few nodes (resecting a minimum of 18 per affected side is advised!) puts the patient at risk and this probably cannot be overcome by administering PD-1 blockers. Without knowledge from randomized controlled trials, nothing should be considered as practice-changing. Moreover, the authors discuss PD-1 blockade extensively despite not showing any link between their research and use of PD-1 blockers in a comparable setting. The authors used ex-vivo stimulation using a not further characterized peptide pools in their experiments. Therefore, and as crude peptide pools are well-known contributors to experimental artifacts, the peptide pool should be characterized (this is so far missing). Moreover, to definitively describe the T cells as specific for E6 or E7 of HPV16, using fluorescent MHC-Tetramers or other HLA loaded with HPV16 E6- or E7-derived peptides would be very helpful. It would be beneficial if the authors can link fluorescent staining of E6 or E7-specific T cells by MHC-Tetramers and production of IFN and/or TNF according to intracellular staining. Without supporting evidence from such experiments (or reliable in-vivo data), only indirect conclusions can be made. Consequently, the paragraph stating the limitations requires an update. Overall, please interpret the findings with more caution as overselling is never a good idea. However, there are aspects in the paper that can be of interest for head and neck surgeons, oncologists and the other members of the interdisciplinary team treating head and neck cancer.
Here are some points to consider during revision:
The title of the paper appears to me inadequate, as neither the immune status as the sum of multiple immunologic characteristics is sufficiently analyzed nor reliable practice-changing recommendations (as I have expected to find within the manuscript by accepting the invitation to review this paper) are substantiated.
Within the Abstract, the detection of E6- and E7-specific T cells in distal and proximal lymph nodes is wrongly stated (reciprocally confused according to their Table 2).
There are inconsistencies in use of abbreviations (INF vs IFN).
Line 67 "immune response of MLN ..." should be replaced by "immune response in MLN..."
Lines 99 to 107 belong to "Discussion" section, not M&M
Line 112: Please correct H2O2 to H2O2
Line 124: "a ×20 objective" should surely be "a 20× objective"
Lines 124 to 130 belong to "Discussion" section, not M&M
Lines 143 to 147 belong to "Discussion" section, not M&M
Lines 151 ff: Please charcterize the oligopeptide pools (OPP) derived from HPV-encoded antigens E6 and E7.
Lines 173 to 174 make a statement about a general characteristic of measurements which do not apply here, as the software does not differentiate T cells "based on their antigen-specific reactivity" but detects patterns of receptors and other proteins (antigens) expressed on the surface of T cells characteristic for particular (functional) T cell subsets. Please revise.
Lines 174 to 178 rather belong to"Results" or "Discussion" section, not M&M
Line 294 please corrected by replacing "tumor-infiltrated" by "tumor-infiltrating".
Lines 301 to 305: I do not share the authors' view that detection of a low frequency of an anti-E6 or anti-E7 immune response in a fraction of CD8+ T cells of one cancer of unknown primary patient indicates the need for changing practice in surgical procedures in head and neck cancer, selective neck dissection in particular. Even if the only E6- and E7-reactive CD8+ cytotoxic T cells of the patient are present in the distant and proximal lymph node of this patient outside the single metastatic lymph node that was altogether with the other lymph nodes resected, the patient will be cured. However, it may indicate that using anti-PD-1 antibodies like pembrolizumab or nivolumab will probably have lower efficacy. And this, indeed, is reflected by the number of negative clinical trials in treatment of HNSCC in the adjuvant setting after curative resection.
Lines 311 to 313 The reference should be provided.
Lines 341 ff in Conclusions is a far-fetching statement without evidence, the authors' opinion at best. The statement, that "the immune function [of T cells] in non-metastatic lymph nodes is important when administering immunotherapy is not supported by data provided by the authors but may be their interpretation as they know about the experiments in mice. The advice given by the authors, that "head and neck surgical oncologists treating HNC should consider the place of immunotherapy and neck dissection in the treatment of head and neck cancer." indeed is what they already do when resecting head and neck cancer: complete removal of the primary malignancy and the lymph nodes at least of the affected neck, whenever there are signs of metastasis or pathological enlargement of lymph nodes, followed by risk factor-adapted adjuvant (postoperative) irradiation. Considering neoadjuvant use of pembrolizumab or other PD-1 blockers would be a way to increase antitumoral immune responses, not sparing resection of potentially affected neck nodes without evidence.
There are some aspects critical for interpretation of findings and their context-dependent discussion missing:
The problematic of sole sentinel node resection (of limited success in maxillofacial surgery, but disastrous in oropharyngeal and hypopharyngeal HNSCC).
Referencing PD-1 blockade without knowing at least something about the effect of PD-1 targeting using pembrolizumab, nivolumab or another anti-PD-1 monoclonal in their experimental setting of stimulating and re-stimulating PBMC ex vivo is out of context, as only activation of T cells to produce cytokines is analyzed. Indeed, it is possible that using PD-1 blockade may have resulted in higher frequencies in activated T cells, but unfortunately, this was not assessed.
Reviewer 2 Report
The article is well written, in clear scientific language. The methodology is very rigorous.
This study had limitations: the authors included only 4 cases and only 2 of the 7 proteins defined as HPV16 virus-derived tumor-associated antigens were used to evaluate the T-cell immune response. Also only in case 1, an immune response was demonstrated by the presence of HPV16 E6-specific and E7-specific T-cells in both the lymphocytes from peripheral blood and the lymph nodes and this case was with unknown primary. The authors explained this aspect by individual immune response to cancer but, maybe it would be necessary a little discussion about the status of the tonsil in this case /if the tonsillectomy had been performed.
The article opens some perspectives in head and neck oncology, both on neck dissection techniques and the potential role of neoadjuvant immunotherapy. No doubt, the immune response to HPV16 antigens in HPV16+ HNSCC requires further evaluation.
I congratulate the authors and my recommendation is for publishing the article after some minor revisions:
Line 73: suppressor instead of ’’sup-presor’’.
Line 102: findings. The selection instead of ’’findings.The selection’’.
Line 297: 77% and 44% of T-cells in PBL instead of ‚’’77% and 44% of PBL’’
Line 474 – The font of Reference 4 needs to be modified.
